# Enhancing the Explainability of Gradient Boosting for Regression Problems through Comparable Samples Selection

## Abstract

Gradient-boosted decision Trees (GBDT) is a highly effective learning method widely used for addressing classification and regression problems. However, akin to other ensemble methods, GBDT suffers from a lack of explainability. Explainability is a desirable property: the ability to discover relationships between input data attributes and the ultimate model predictions is crucial for a comprehensive understanding of the GBDT method. To enhance the explainability of such algorithms, we propose to exhibit particular training data, referred to as *comparable samples*, upon which the model heavily relies for specific predictions. To that end, we show that a prediction of GBDT can be decomposed as a weighted sum of training data when using specific loss functions. It is noteworthy that these weights may be negative. Furthermore, during the prediction of a training sample's response, the weights associated with other training samples in the prediction's decomposition vanish, indicating a potential issue of overfitting. To overcome this issue, we introduce nonnegativity constraints on the weights and substitute gradient descent with a methodology inspired by the Frank-Wolfe algorithm called Explainable Gradient Boosting (ExpGB). The predictions generated by the proposed algorithm can be directly interpreted as convex combinations of the training targets. This allows for selecting training data resembling a given sample by comparing their decomposition coefficients. We conduct a comparative analysis with classical GBDT algorithms across diverse datasets to validate the estimation quality. Additionally, we evaluate the fidelity of comparable samples by demonstrating their proficiency in estimating the characteristics of the considered sample. Our approach, thus, offers a promising avenue for enhancing the explainability of GBDT and similar ensemble methods.

## 1 Introduction

Historically, machine learning techniques were designed to perform well on accuracy metrics, placing less emphasis on other criteria, such as explainability. However, explainability enhances trust, and users are more likely to use a specific model if they understand how a prediction was made. In particular applications, basic models such as linear regression may be preferred over more accurate but complex ones. On top of that, model explainability can help assess if a model behaves as expected before deploying it in real-world applications where data may differ significantly from validation datasets. Many techniques were developed in the explainable artificial intelligence domain (XAI) to enhance human understanding of machine learning models (Adadi & Berrada, 2018). The literature defines several notions, such as understandability, comprehensibility, interpretability, and transparency (Arrieta et al., 2020). Explainability is usually defined as an interface between users and models that is understandable to human users and an accurate proxy of the model.

Several methods and tools have been proposed to enhance the explainability of ML models. Feature importance (Breiman, 2001; Ishwaran, 2007) was initially introduced to facilitate feature selection but is now widely used to explain any tree ensemble model. It aims to quantify each input feature's contribution to the global predictions. LIME (Ribeiro et al., 2016) explains any classifier or regression model individual prediction by learning an interpretable model around this local prediction. SHAP values (Lundberg & Lee, 2017; Lundberg et al., 2020) enhance a model's explainability by approximating the model with an additive

interpretable model. With this approach, one can understand the effect of each variable on a given prediction. These methods are model-agnostic but can hardly take the specificity of each model into account and are often expensive to compute for large datasets.

In this work, we focus on regression problems and aim to explain a prediction by extracting data from the training set with similar features and similar responses to a given member of the testing set. This way of explaining or interpreting a prediction may be preferred to methods that create summaries of features, such as the methods cited above, especially for non-expert users. This is particularly the case when an instance of the data can be represented in a humanly understandable way. For instance, in real estate estimation problems, this definition is more similar to the analyses performed by real estate agents to evaluate the price of a property.

In particular, this paper is concerned with Gradient Boosted Decision Trees (GBDT)(Friedman, 2001; 2002; Hastie et al., 2009), a prevalent method proven to perform well on a wide range of regression and classification problems (Caruana & Niculescu-Mizil, 2006; Grinsztajn et al., 2022). In recent years, GBDT algorithms have improved to achieve better estimation performances (Prokhorenkova et al., 2018; Chen et al., 2015), without improving explainability. In parallel, methods have been developed to make gradient boosting more interpretable, for instance, with boosted generalized additive models (Lou et al., 2013). Still, these methods remain less efficient as they only consider the interaction between pairs of features and not the entire interaction between features like trees ensemble. More recently, there have been some improvements in GBDT explainability, for example, in (Delgado-Panadero et al., 2022). However, these works usually focus on feature-based explanations, while we aim to develop an example-based explanation.

The problem considered here is similar to the prototype selection problem, which was extensively studied for classification problems(Tan et al., 2020; Bien & Tibshirani, 2011; Kim et al., 2014; Gurumoorthy et al., 2017; 2019), but remains relatively unexplored for regression problems (Arnaiz-González et al., 2016; Kordos & Blachnik, 2012). One reason is that members of a given class are homogeneous, while samples yielding similar responses can be very diverse in regression. Consequently, the method proposed here is local because the chosen representative training samples, *comparable sample*, will depend on the sample of interest.

## 1.1 Contribution

The main contribution of the paper is the introduction of a variant of GBDT for regression problems, aiming at improving the explainability of the results. To this end, the method extracts training samples similar to a given test sample, both in terms of features and response.

The proposed method is based on the observation that in GBDT-applied-to-regression problems with the $\ell_2$ loss, a prediction can be expressed as a linear combination of target values of the training dataset. Intuitively, training samples that appear with high weights in the decomposition of the predicted response of a training sample will be considered similar to the training sample. A limitation of this approach is that these coefficients are only computable for specific loss functions, are not guaranteed to be nonnegative, involve many training values for each prediction, and are expensive to compute.

To address these issues, we propose to constrain the learned model to decompose its predictions as linear combinations of the responses of the training set, *with positive weights*. Based on the Frank-Wolfe algorithm, a variant of the gradient boosting algorithm is proposed to account for these additional constraints. We show that the induced algorithm fixes the previously mentioned issues, and that it is can be applied to more general loss functions. This algorithm has an additional advantage: the number of iterations determines the upper bound for nonzero weights. Consequently, the resulting weight vector exhibits sparsity.

This approach of modifying the gradient algorithm was investigated to accelerate GBDT following the principles of Nesterov's descent (Biau et al., 2019), with application to massive amounts of high-dimensional data, or by using the Frank-Wolfe algorithm to solve a $\ell_1$ penalized problem to reduce overfitting (Wang et al., 2015).

In addition to providing a prediction, our approach gives the weights used to attain this prediction as a weighted sum of the responses of the training data. A measure of similarity between samples is computed

from these weights, assuming that similar samples will yield similar decomposition for the prediction of their response.

An application of this measure of similarity is the selection of a set of training samples similar to the sample of interest. This set of samples can then be used to explain the prediction.

Numerical tests conducted on real data show that the proposed method provides similar prediction performance as the best state-of-the-art GBDT algorithms: catboost (Prokhorenkova et al., 2018), XGBOOST (Chen et al., 2015), and the scikit-learn implementation of gradient boosting (Pedregosa et al., 2011). It is also shown by comparing decomposition weights that comparable samples to a given sample can be extracted from the training or testing database. The ensuing comparable samples are very close to the considered sample regarding features and response. Moreover, decomposition weights obtained by GBDT methods for the prediction of a member of the training set involve only this particular sample at high iteration numbers, which indicates overfitting. Decomposition weights for the proposed method remain diverse when the number of iterations increases.

This article recalls first, in section 2, the gradient boosting algorithm and, in the case of the $\ell_2$ loss, interprets its prediction as linear combinations of the training data. The modifications of the algorithm improving the explainability are introduced in section 3, for more general losses. Section 4 gives results obtained on various real-world datasets, using the $\ell_2$ and $\ell_1$ losses. Section 5 concludes this paper.

## 2 Gradient tree boosting

In regression problems, one aims at learning a function $F$ mapping features $\mathbf{x} \in \mathbf{R}^d$ to a response $y \in \mathbf{R}$, so that it minimizes the expected loss $\mathcal{L}(F) = E(L(y, F(\mathbf{x})))$ where $L$ denotes a loss function, and $E$ is the mathematical expectation. This function $F$ is trained using $N$ training samples $\{(\mathbf{x}_n, y_n)\}$, assumed to be distributed according to the joint distribution $p(\mathbf{x}, y)$, by minimizing the empirical expectation of the loss

$$\mathcal{E}_F = \frac{1}{N} \sum_{n=1}^{N} L(y_n, F(\mathbf{x}_n)) \tag{1}$$

$$\equiv l(\mathbf{f}), \tag{2}$$

where

$$\mathbf{f} = (F(\mathbf{x}_n))_n \tag{3}$$

is the vector of $\mathbf{R}^N$ collecting the estimated responses of the training samples $\mathbf{x}_n$.

As the amount of available data is limited, the minimizers of Eq. (1) are not guaranteed to yield accurate estimations of $F(\mathbf{x})$ for $\mathbf{x}$ not in the training dataset. Introducing priors on $F$ can alleviate this problem. Usual choices include linear models, regularity assumptions, covariance models (e.g. used in kriging interpolation), or parametric models, where a small number of parameters can describe $F$. Gradient tree boosting follows the latter approach. More specifically $F$ is restricted to belong to the set : $\left\{ F, F(\mathbf{x}) = \sum_{t=1}^{T} \gamma_t h_{\theta_t}(\mathbf{x}) \right\}$, where the $h_{\theta_t}$ are simple functions parameterized by low-dimensional parameters $\theta_t \in \Theta$, where $\Theta$ is a low-dimensional parameter space, and the $\gamma_t$ are real weights. We assume that the set of simple functions is stable by multiplication by a scalar, that is for any $\theta$ and scalar $\alpha$, there exists $\theta^\star$ such that $\alpha h_\theta = h_{\theta^\star}$.

Gradient boosting is an iterative algorithm, which generates a sequence of functions $F_t$, inspired by gradient descent: at each iteration $t$, an update $h_{\theta_t}$ is added to decrease the empirical loss $\mathcal{E}_{F_t}$. This update is selected so that its values $\mathbf{h}_{\theta_t} \in \mathbf{R}^N$ on the training points $\mathbf{x}_n$ is the most parallel with the gradient $\mathbf{g}_{t-1} = \operatorname{grad} l(\mathbf{f}_{t-1})$ for $\mathbf{f}_{t-1} = (F_{t-1}(\mathbf{x}_n))_n$ of the empirical loss at the previous iteration, and is found by solving the optimization problem

$$\theta_t = \underset{\theta \in \Theta}{\arg\min} \, \|\mathbf{g}_{t-1} - \mathbf{h}_\theta\|_2^2. \tag{4}$$

The fact that $\mathbf{h}_{\theta_t}$ is the most parallel to $\mathbf{g}_{t-1}$ can be seen by replacing this optimization problem by the equivalent problem

$$(\alpha^\star, \theta^\star) = \underset{\alpha \in \mathbf{R}, \theta \in \Theta_n}{\arg\min} \ \|\mathbf{g}_{t-1} - \alpha \mathbf{h}_\theta\|_2^2 \,, \tag{5}$$

where $\Theta_n$ is the subset of $\Theta$ such that $\|\mathbf{h}_\theta\|_2 = 1$. Rewriting the objective function as $\|\mathbf{g}_{t-1}\|_2^2 - \alpha \langle \mathbf{g}_{t-1}, \mathbf{h}_\theta \rangle + \alpha^2$, the optimal $\mathbf{h}_{\theta^\star}$ maximizes the scalar product with $\mathbf{g}_{t-1}$, or equivalently, minimizes the angle with $\mathbf{g}_{t-1}$. As $\mathbf{h}_{\theta_t}$ and $\mathbf{h}_{\theta^\star}$ are co-linear, $\mathbf{h}_{\theta_t}$ also minimizes the angle with $\mathbf{g}_{t-1}$.

It is worth observing that Eq. (5) involves the minimization of an $\ell_2$ norm, independently of the choice of the loss $L$. Once $\mathbf{h}_{\theta_t}$ is fitted, a line search is conducted similarly as in the steepest descent strategy:

$$\gamma_t = \underset{\gamma}{\arg\min} \ l(F_{t-1} + \gamma h_{\theta_t}) \tag{6}$$

$$= \underset{\gamma}{\arg\min} \ \frac{1}{N} \sum_{n=1}^{N} L(y_n, F_{t_1}(\mathbf{x}_n) + \gamma h_{\theta_t}(\mathbf{x}_n)) \,. \tag{7}$$

---

**Algorithm 1** Gradient boosting

---

**Input:** $\{(\mathbf{x}_n, y_n)\}_{n=1,\ldots,N}$, T,
  $F_0 = \arg\min_C \mathcal{E}_C = \arg\min_C \sum_{n=1}^{N} L(y_n, C)$
  **for** t = 1 to T **do**
    Compute $\mathbf{g}_{t-1} = \text{grad} \ l(\mathbf{f}_{t-1})$
    $\theta_t, \alpha_t = \arg\min_{\alpha \in \mathbf{R}, \theta \in \Theta} \|\mathbf{g}_{t-1} - \alpha \mathbf{h}_\theta\|_2^2$
    $\gamma_t = \arg\min_\gamma \mathcal{E}_{F_{t-1}} + \gamma \mathbf{h}_{\theta_t}$
    Update $F_t(\mathbf{x}) = F_{t-1}(\mathbf{x}) + \gamma_t h_{\theta_t}(\mathbf{x}))$
  **end for**

---

A popular choice for $h_{\theta_t}$ is to use shallow regression trees (Breiman et al., 1993), $\theta_t$ corresponding in this case to the splitting features, splitting location, and terminal node values of the tree. In practice, solving (5) simply corresponds to training a tree on the set $\{(\mathbf{x}_n, g_{t-1,n})\}$ with quadratic loss, where $g_{t-1,n}$ is the $n$-th coordinate of $\mathbf{g}_{t-1}$.

Such trees can be described by subsets $\{A_k^t\}$ covering the space, and values $b_k^t$, with

$$h_{\theta_t} = \sum_{k=1}^{K} b_k^t \mathbf{1}_{A_k^t} \,, \tag{8}$$

and

$$b_k^t = \frac{1}{\#\{n, \mathbf{x}_n \in A_k^t\}} \sum_{n \in \{n, \mathbf{x}_n \in A_k^t\}} g_{t-1,n} \,. \tag{9}$$

The number of subsets $K$ depends on the depth of the trees.

Overfitting is avoided by limiting the complexity of each $h_{\theta_t}$ (e.g. by constraining the depth of the regression trees), using a finite number of iterations, and dampening the iterations by setting $F_t(\mathbf{x}) = F_{t-1}(\mathbf{x}) + \lambda \gamma_t h(\mathbf{x}, \theta_t)$ with $0 < \lambda < 1$. Further regularizations can also be applied.

After $T$ iterations, the prediction model $F_T$ is described by the coefficients $\gamma_t$, the sets $A_k^t$ and the values $b_k^t$, $t$ varying from 1 to $T$. The cost of a prediction is the computational cost of applying the weak learners $h_t$ to $\mathbf{x}$.

## 2.1 Unfolding gradient boosting

The proposed method is based on the observation that the estimation of a response by GBDT algorithms, in the case of regression with a $\ell_2$ loss, can be interpreted as a weighted average of responses $y_n$ from the

training data, with weights depending only on the sets $A_k^t$ and the features $\mathbf{x}$. In the case of the $\ell_2$ loss, the function $\mathcal{E}_F$ to minimize corresponds to $\frac{1}{N}\sum_{n=1}^{N}(y_n - F(\mathbf{x}_n))^2$ and thus $\mathbf{g} = 2(y_n - F(\mathbf{x}_n))_n$.

Choosing $F_0 = \frac{1}{N}\sum_{n=1}^{N} y_n$, and from the equations (8), (9) and the update step of Algorithm 1, it is clear, by induction, that a prediction $F_t(\mathbf{x})$ can be obtained as a weighted sum of the training values $y_n$:

$$F_t(\mathbf{x}) = \sum_{n=1}^{N} w_n^t(\mathbf{x}) y_n \,, \tag{10}$$

and that the sum of the weights is equal to 1. Indeed, assuming that $F_{t-1}(\mathbf{x}) = \sum_{n=0}^{N} w_n^{t-1}(\mathbf{x}) y_n$ with $\sum_{n=0}^{N} w_n^{t-1}(\mathbf{x}) = 1$, and observing that $g_t(\mathbf{x}_i) = y_i - F_{t-1}(\mathbf{x}_i)$, each $g_t(\mathbf{x}_i)$ can be decomposed as a linear combination of the $y_n$, with weights summing to 0, as also do the $b_k^t$. Thus, the update conserves the sum of the weights.

In the prediction phase, in addition to $F_t(\mathbf{x})$, weights $w_n^t(\mathbf{x})$ can be computed by the following iterations:

$$w_n^t(\mathbf{x}) = w_n^{t-1}(\mathbf{x}) + \gamma_t \frac{1}{\#\{n', \mathbf{x}_n' \in A_k^t\}} \sum_{n' \in \{n', \mathbf{x}_n' \in A_k^t\}} \delta_{nn'} - w_n^{t-1}(\mathbf{x}_n') \,, \tag{11}$$

with $k$ such that $\mathbf{x} \in A_k^t$, revealing the structure of the estimated value in function of the training values. $\delta_{nn'}$ is defined as 1 if $n = n'$, and 0 if $n \neq n'$.

However, this approach is limited by three main aspects. Firstly, the computation of the weights $w_n^t(\mathbf{x})$ is expensive in time and space. Indeed, the update of the weights at iteration $t$ involves $b_k^t$, which itself involves the prediction of each $\mathbf{x}_n$ falling into the same leaf $A_k^t$ as $\mathbf{x}$. Thus, the weights $w_n^t(\mathbf{x})$ depend on the weights $w_{n'}^{t-1}(\mathbf{x}_{n'})$ of all training vectors $\mathbf{x}_{n'}$ falling into the same leaf $A_k^t$. These weights $w_{n'}^{t-1}(\mathbf{x}_{n'})$ depend on the weights of all training data falling in the same leaf $A_k^t$ as the previously considered data, etc. Ultimately, to obtain the weights $w_n^t(\mathbf{x})$, it is necessary to compute and store the weights of the training data at each iteration, with memory cost $N^2 T$. The cost of computing the decomposition of a prediction is therefore high compared to computing the prediction itself (which only depends linearly on $T$, and is independent of $N$).

Secondly, this decomposition has been obtained in the case of the $\ell_2$ loss, and is unlikely to be easily extensible to other losses: the values $g_t(\mathbf{x}_n)$ must be decomposed as a linear combination of the target values.

Finally, numerical experiments show that the weights $w_n^t(\mathbf{x})$ are not guaranteed to be non-negative, which weakens the explainability of the gradient boosting regression. Similar issues also arise in kriging regression (Deutsch, 1996; Barnes & Johnson, 1984). Furthermore, for a training sample $\mathbf{x}_n'$, it was observed that at high number of iterations $T$, the decomposition weights $w_n^T(\mathbf{x}_n')$ of the estimated output $F_T(\mathbf{x}_n')$ tend towards $\delta_{nn'}$. That is, the estimation of its output, at large $T$, does not involve the other training data, indicating overfitting. Numerical results supporting these observations will be given in section 4.

## 3 Explainable gradient boosting predictions

The previous sections showed that gradient boosting with the $\ell_2$ loss, in addition to an estimation of a response to test data, can also provide the weights used to form this prediction, however with limitations.

To alleviate these limitations, we propose modifying the gradient-boosting algorithm to ensure that the weights $w_n^t(\mathbf{x})$ remain positive, more precisely, to constrain the estimators $F_t$ in the set of functions $\Omega$ where for each $\mathbf{x}$, the weights $w_n^t(\mathbf{x})$ belong to the unit simplex $\Delta = \{(w_1, \ldots, w_N), w_n \geq 0, \sum_{n=1}^{N} w_n = 1\}$. The constrained problem could be theoretically solved by adapting the gradient boosting similarly to the projected gradient algorithm, by projecting $F_t$ in $\Omega$ after each gradient step. Although projection on the unit simplex is a low complexity operation (Condat, 2016), one has to project a set of weights in $\Delta$ for each possible set of weights, that is, for each combination of leaves of the regression tree at each iteration, and for each training or testing sample.

To avoid the combinatorial growth of the number of projections, we suggest following the principles of the Frank-Wolfe algorithm, or conditional gradient algorithm (Frank & Wolfe, 1956; Jaggi, 2013), given in

Algorithm 2. At each iteration, the gradient of the objective function is computed, and the minimizer $s$ of the linear approximation of the objective function is searched. Then, the new iterate is found as a convex combination of the previous iterate and the minimizer of the linearized problem. As the feasible set is convex, the iterates are guaranteed to satisfy the constraints, and no projection is needed. Furthermore, the solution $s$ of the linearized problem is frequently easily obtained. Application of the Frank-Wolfe algorithm to regularized gradient boosting was investigated in (Wang et al., 2015), and similarities between the Frank-Wolfe algorithm and boosting variational inference were highlighted in (Locatello et al., 2018).

We consider the scalar product $\langle \cdot, \cdot \rangle$ defined by

$$\langle s, h \rangle = \int_{\mathbf{R}^d} s(\mathbf{x})h(\mathbf{x})p(\mathbf{x})d\mathbf{x}\,, \tag{12}$$

where $p$ is the probability density of $\mathbf{x}$.

In our algorithm, the tree $h_{\theta_t}$ is replaced by $s_t$, obtained by solving the problem

$$s_t = \arg\min_{s\in\Omega_m} \langle s, h_{\theta_t} \rangle \,, \tag{13}$$

where $\Omega_m$ is the set of functions of $\Omega$, such that the weights $w_n^t(\mathbf{x})$ for a $\mathbf{x}$ in $A_k^t$ involve training samples in $A_k^t$. This choice is made to ensure that the weights $w_n^t(\mathbf{x})$ are increased only for training samples $\mathbf{x}_n$ falling in the same leaf as the tested sample.

Expliciting $h_{\theta_t}$ and writing $s(\mathbf{x}) = \sum_{n=1}^{N} w_n(\mathbf{x})y_n$,

$$\langle s, h_t \rangle = \int_{\mathbf{R}^d} \sum_{n=1}^{N} w_n(\mathbf{x})y_n \sum_{k=1}^{K} b_k^t \mathbf{1}_{A_k^t}(\mathbf{x})p(\mathbf{x})d\mathbf{x}\,. \tag{14}$$

This scalar product is maximized by maximizing the integrand pointwise, which does not necessitate the probability density $p$. For a given $\mathbf{x}$ falling in the leaf $A_{k^\star}^t$, the integrand

$$\sum_{n=1}^{N} w_n(\mathbf{x})y_n b_{k^\star}^t \,, \tag{15}$$

is maximized by setting $w_n(\mathbf{x}) = 1$ for the index $n_k^t = n^+$ or $n^-$ depending on the sign of $b_{k^\star}^t$, with $n^+$ and $n^-$ the index of the largest, resp. smallest, $y_n$ in $A_{k^\star}^t$.

The complete algorithm is given in Algorithm 3. In addition to the fact that, for a given $\mathbf{x}$, the weights are guaranteed to be positive and to sum to one, the update step shows that at each iteration for a given $\mathbf{x}$, at most one new nonzero weight is added to the convex combination. Consequently, storing the index $n_k^t$, (either $n^+$ or $n^-$) for each set $A_k^t$ at each iteration is sufficient to compute the decomposition weights.

The prediction algorithm is given in Algorithm 4, yielding the predicted response, and its decomposition weights as a sum over the training responses. It is worth noting that the proposed algorithm is not strictly a Frank-Wolfe algorithm. Indeed, a regression tree is used instead of the gradient, and $s$ is searched in a smaller space than the feasible set $\Omega$.

---

**Algorithm 2** Frank-Wolfe Algorithm

---

$x^0 \in \Omega$
**for** t = 1 to T **do**
    Compute $s := \arg\min_{s\in\Omega}\langle s, \nabla f(x^t)\rangle$
    $\gamma := \frac{2}{t+2}$
    Update $x^{t+1} = (1-\gamma)x^t + \gamma s$
**end for**

---

---

**Algorithm 3** ExpGB, fit

---

**Input:** $\{(\mathbf{x}_n, y_n)\}_{n=1,\ldots,N}$, T,

    initialization : $F_0(\mathbf{x}_n) = \frac{1}{N}\sum_{n=1}^{N} y_n, n = 1\ldots N$

    **for** t = 1 to T **do**

        Compute $\mathbf{g}_{t-1} = \text{grad } l(\mathbf{f}_{t-1})$

        Fit a tree $h_{\theta_t}$ on the data $\{(\mathbf{x}_n, \mathbf{g}_{t-1})\}, n = 1\ldots N$ with leaves $A_k^t$ and values $b_k^t$

        Set $n_k^t = \begin{cases} \arg\max_{\{n, \mathbf{x}_n \in A_k^t\}} y_n & \text{if } b_k^t > 0 \\ \arg\min_{\{n, \mathbf{x}_n \in A_k^t\}} y_n & \text{else} \end{cases}$

        $\gamma := \frac{2}{t+2}$

        $F_t(\mathbf{x}_n) = (1-\gamma)F_{t-1}(\mathbf{x}_n) + \gamma y_{n_k^t}$, where $\mathbf{x}_n \in A_k^t$

    **end for**

    **return** $h_{\theta_t}, t = 1\ldots T$

---

**Algorithm 4** ExpGB, predict

---

**Input:** $\mathbf{x}$, trees $h^t$, indices $n_k^t$, values $y_n$

    $F_0(\mathbf{x}) = \frac{1}{N}\sum_{n=1}^{N} y_n$

    $w_n^t(\mathbf{x}) = 1/N$

    **for** t = 1 to T **do**

        $\gamma := \frac{2}{t+2}$

        With $k$ such that $\mathbf{x} \in A_k^t$

        $F_t(\mathbf{x}) = (1-\gamma)F_{t-1}(\mathbf{x}) + \gamma y_{n_k^t}$

        $w_n^t(\mathbf{x}) = (1-\gamma)w_n^{t-1}(\mathbf{x}) + \gamma \delta_{nn_k^t}$

    **end for**

---

### 3.1 Measure of similarity

We now introduce a measure of similarity between samples based on the decomposition weights of the prediction of their response. An application of this measure of similarity is to find training data similar to a testing sample in the sense that they have similar weights and thus are part of the same leaves in a high number of iterations.

The similarity between two points $\mathbf{x}$ and $\mathbf{z}$ is assessed by comparing the decomposition weights $\mathbf{w}^T(\mathbf{x})$ and $\mathbf{w}^T(\mathbf{z})$, e.g. by computing their $\ell_1$ distance. We define the distance

$$d(\mathbf{x}, \mathbf{z}) = \|\mathbf{w}^T(\mathbf{x}) - \mathbf{w}^T(\mathbf{z})\|_1 \,, \tag{16}$$

where $\mathbf{w}^T(\mathbf{x}) = (w_1^T(\mathbf{x}), \ldots, w_N^T(\mathbf{x}))$. The following observations support this choice:

- a small distance $d(\mathbf{x}, \mathbf{z})$ implies that the estimated response is similar. Indeed, $|F_T(\mathbf{x}) - F_T(\mathbf{z})| \leq d(\mathbf{x}, \mathbf{z}) \max |y_n|$. Conversely, a small change in response does not imply small changes in features. This is desirable, as similar responses do not necessarily imply similar features.

- samples frequently falling in the same leaves will have similar decomposition weights and, therefore, the distance between them will be small.

A set of members of the training set similar to a testing sample is then obtained by picking the $K$ closest samples (with $K$ user-defined) or by picking samples whose distance to the considered testing sample is below a given threshold.

We note that gradient boosting does not allow such comparison. Indeed, training samples, even though similar, will always have a mutual distance $d$ tending to 2, as their weights concentrate on themselves.

| Dataset_name | n_sample | n_feature |
|---|---|---|
| Abalone | 4177 | 8 |
| Ailerons | 13750 | 33 |
| Bike_sharing_demand | 17379 | 6 |
| Brazilian_houses | 10692 | 8 |
| CPU_act | 8192 | 21 |
| Diamonds | 53940 | 6 |
| Elevators | 16599 | 16 |
| Houses | 20640 | 8 |
| House_16h | 22784 | 18 |
| House_sales | 21613 | 15 |
| Medical_charges | 163065 | 3 |
| Miami_housing_2016 | 13932 | 14 |
| NYC_taxi_green_dec_2016 | 581835 | 9 |
| Pol | 15000 | 26 |
| Sulfur | 10081 | 5 |
| Superconductor | 21263 | 79 |
| Wine | 6497 | 11 |
| Yprop_4_1 | 8885 | 61 |

Table 1: Number of observations and features in each dataset

Because the similarity measure introduced here consists of a $\ell_1$ distance between two weight vectors, it is not necessary to store the weights coming from $F_0$ as they are the same for all observations. Hence we can consider that the weight vectors are sparse, with a maximum of $T$ non-null coefficients.

## 4 Results

### 4.1 Datasets

To evaluate the performance of the proposed approach, we will compare it to other gradient boosting methods - Catboost, XGBOOST (XGB), and the scikit-learn implementation of gradient boosting - on several classic public datasets with regression tasks from OpenML (Vanschoren et al., 2014). More precisely, our experiment setup consists of 18 regression datasets as defined in (Grinsztajn et al., 2022), that are compiled in Table 1. These datasets contain various problems with tabular data that enable these estimators to be properly evaluated: they have heterogeneous columns, have no missing values, are not high dimensional, are well documented, come from real-world problems, are not too small, are not too easy, and are not deterministic. In addition, the choice of data sets covers a wide variety of cases, with the number of data ranging from 4k to 580k and the number of variables from 3 to 79.

### 4.2 Evaluating the models

To assess the models, we randomly split the data into a training dataset representing 75% of the data and a test dataset with the remaining 25%. The models are trained using the best hyperparameters selected as described in section 4.2.1.

### 4.2.1 Hyperparameter selection

We considered two parameters to be optimized: the number of iterations (200,400,600,800,1000) and the depth of the trees (2,3,4,5,6,7,8,9,10). To select the hyperparameters of each model, we used a standard approach by taking a grid search with a K-fold cross-validation (K being 5 in most cases) on the training dataset. That is, we split the training data into 5 folds and trained the models on 4 of them while assessing the result on the last one. We then repeated the process 4 times so that each fold was used exactly once as

| Dataset | Cat | XGB | Sklearn | ExpGB |
|---|---|---|---|---|
| *Abalone* | 4.46e0 | 4.55e0 | 4.45e0 | **4.33e0** |
| *Ailerons* | **2.26e-8** | 2.56e-8 | 2.39e-8 | **2.26e-8** |
| *Bike_sharing_demand* | **1.29e3** | 1.55e3 | 1.53e3 | 1.40e3 |
| *Brazilian_houses* | **1.28e-1** | 1.30e-1 | 1.32e-1 | **1.28e-1** |
| *CPU_act* | 4.94e0 | 4.82e0 | 4.94e0 | **4.77e0** |
| *Diamonds* | 2.76e5 | 2.97e5 | 2.85e5 | **2.67e5** |
| *Elevators* | **3.72e-6** | 3.99e-6 | 3.94e-6 | 4.01e-6 |
| *Houses* | 4.61e-2 | 5.05e-2 | 5.07e-2 | **4.54e-2** |
| *House_16h* | 3.67e-1 | 3.75e-1 | 4.05e-1 | **3.59e-1** |
| *House_sales* | 2.74e-2 | 2.85e-2 | 2.88e-2 | **2.68e-2** |
| *Medical_charges* | 6.56e-3 | 6.48e-3 | 6.55e-3 | **6.44e-3** |
| *Miami_housing_2016* | 1.95e-2 | 2.26e-2 | 2.14e-2 | **1.94e-2** |
| *NYC_taxi_green_dec_2016* | **6.50e0** | 6.52e0 | 6.51e0 | 6.65e0 |
| *Pol* | **1.64e1** | 2.39e1 | 2.54e1 | 1.72e1 |
| *Sulfur* | **9.17e-4** | 1.19e-3 | 1.13e-3 | 1.13e-3 |
| *Superconductor* | **8.24e1** | 8.44e1 | 8.70e1 | 9.03e1 |
| *Wine* | **3.49e-1** | 3.70e-1 | 3.72e-1 | 3.79e-1 |
| *Yprop_4_1* | **7.20e-4** | 7.46e-4 | 7.43e-4 | 7.27e-4 |

Table 2: Errors (MSE) on the test set for each of the 18 datasets tested using Catboost, XGboost, the implementation of gradient boosting in scikit learn, and ExpGB using a $\ell_2$ loss function. All of the models are trained using hyper-parameters obtained with grid-search which depend on the dataset. For each dataset, the best performance is in bold.

a validation dataset. We selected the hyperparameters that minimize the average of the mean squared error on the 5 test subsets.

### 4.2.2 Data preprocessing

We used as few data preprocessing processes as possible and only some classical transformations, which are:

- Remove feature: We removed features in some datasets when their variance was very low. For instance, in the elevator dataset we removed diffSaTime2 and diffSaTime4, as their standard variations were around $1e-6$.

- Dummies: Categorical variables are encoded as dummy features for all models.

- Log transformation of target feature: for datasets having heavy-tailed target features, we log-transform them. This concerns the following datasets: *diamonds*, *houses*, *house_16H*, *house_sales*, *medical_charges*, *MiamiHousing2016*, *nyc_taxi_green_dec_2016*.

### 4.3 Prediction performances

The prediction performance of ExpGB is compared with three reference models (Catboost, XGBoost, Sklearn) for each of the 18 datasets. More precisely, Table 2 shows the MSE on each test set for the three reference models and ExpGB. As expected the three models reach similar performances on all datasets. ExpGB exhibits similar performances as well, proving the modifications made to enhance explainability are not done at the expense of a decrease in performance. In particular, we reach a minimal MSE with ExpGB for ten datasets and the same amount for Catboost. The thin margins prevent us from concluding that ExpGB outperformed the most popular implementations of GBDT but these results indicate that we can expect comparable performances on any given datasets. The analysis of the results reveals another interesting property of our new implementation of GBDT. Figure 1 plots the MSE of all four model implementations on the train set (upper panel) and the test set (lower panel) as a function of the number of iterations for

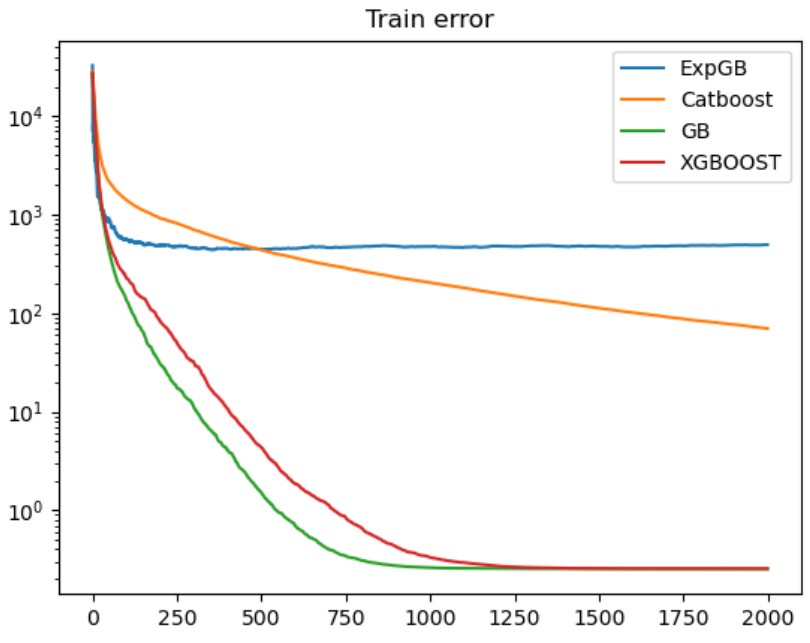

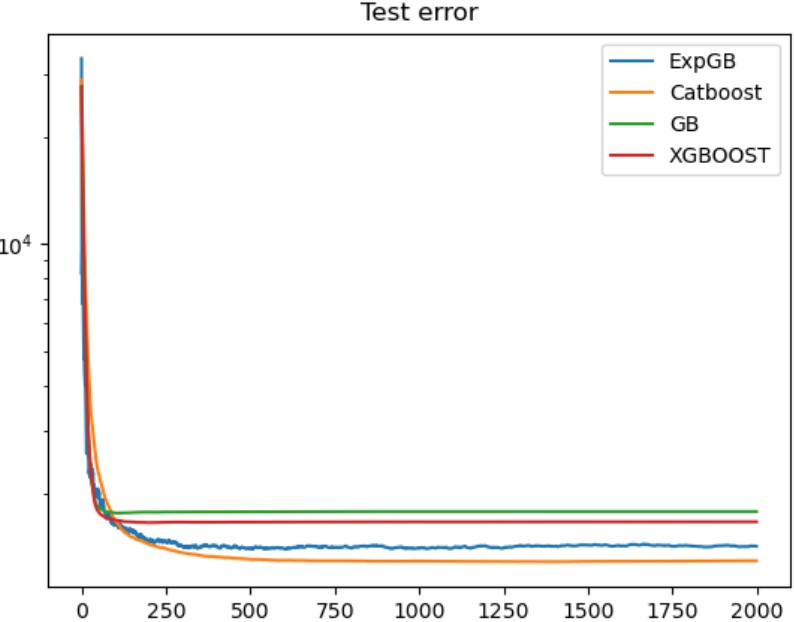

Figure 1: Bike-sharing demand dataset. Mean squared error on both the training dataset and the test dataset for each tested model as a function of the number of iterations.

the bike-sharing dataset. For the visualization, the depth chosen is the same for all models, i.e. the optimal depth for catboost obtained using cross-validation. It appears that ExpGB overfits much less than the other gradient-boosting algorithms. Although we present these plots only for the case of bike sharing, this property holds for all other datasets. An intuition of this apparent immunity against over-fitting is proposed in 4.6.

### 4.4 Extracted prototypes at a glance

The main objective of our algorithm ExpGB was to enhance the explainability of GBDT by extracting similar training samples to a given tested sample. Here we are looking at these prototypes through examples extracted from the house-sales dataset. This dataset aims to predict the sale price of different houses in King County located in the USA. It contains information about the houses such as their location, their characteristics (e.g. number of bathrooms, area,...), and their sale date. We chose to explore this dataset because it is common to consider similar houses in real estate regression problems. Figure 2, for a given testing sample, similar training samples according to three different similarity measures on each row. Each column represents a feature, the last being the price to be estimated. The represented features are selected by decreasing feature importance. In the last column, the purple square indicates the estimated response.

In each panel is represented the histogram of the feature in the training dataset (in grey), the value of the feature for the tested sample (green arrow), and the values of the feature of the ten most similar training samples (blue disk, the higher the more similar).

The first row represents the prototypes extracted using our method as described in section 3.1. We can observe that, for the four most important features, the samples extracted are very close in terms of features even though the testing sample's features are in the tail of each feature distribution. Also, the responses of the extracted prototypes (last column) are close to the attained sell price of the tested sample and its estimation.

The second row represents the training samples with the sell price being the closest to the estimated price of the testing sample. We can easily see that these samples are very different in terms of features, meaning that the task of identifying similar samples is not trivial.

Finally, the last row of this figure corresponds to the closest training samples in terms of the Mahalanobis distance of their features. The obtained training samples are not as similar as for the proposed method, in particular for the second and third features.

It is well known that real estate prices are highly dependent on the location of the property. Figure 3 shows, on two different houses with exactly the same sale price (500 000$), that the proposed method can capture the importance of location. We can see from this example that in regression problems, prototypes must depend on the features and not only on the response of the observations, implying a very local approach.

### 4.5 Assessing systematic prototype relevance

In the previous sections, we defined (3.1) and showed a telling example of a real-life use case (4.4) with the similar samples ExpGB provides on top of the regression result. Here we intend to prove they are systematically relevant, not only by cherry-picking examples. To be meaningful, prototypes should exhibit several properties. First, the response value of the most similar training samples has to be close to the response of the chosen sample. On top of that, for the users to consider them as relevant and comparable to their inputs, we would like them to have similar features. This is especially true for non-expert users. Finally, similar samples extracted with the method must be stable at high iterations.

#### 4.5.1 Similarity of prototypes response

We introduce a $K$ nearest neighbours (KNN ExpGB) estimator based on the proposed similarity distance in section 3.1. We emphasize here that this estimator is not intended to be used in practice, but highlights the ability of the proposed similarity measure to identify prototypes similar to a testing sample in response. The response of a testing sample is obtained by averaging the responses of the $K$ (here, $K = 10$) closest training samples. The number of prototypes $K$ used for KNN ExpGB was set arbitrarily to 10, which is low enough to remain intelligible to the user, but high enough to get sufficient information. MSE of this estimator is compared with ExpGB in Table 3, with the variance of the responses (i.e., MSE of the estimation of the response by the mean of the complete training set), and with the MSE of a classic KNN trained using the euclidean distance between normalized features (KNN euclidean). The number of neighbours defined with

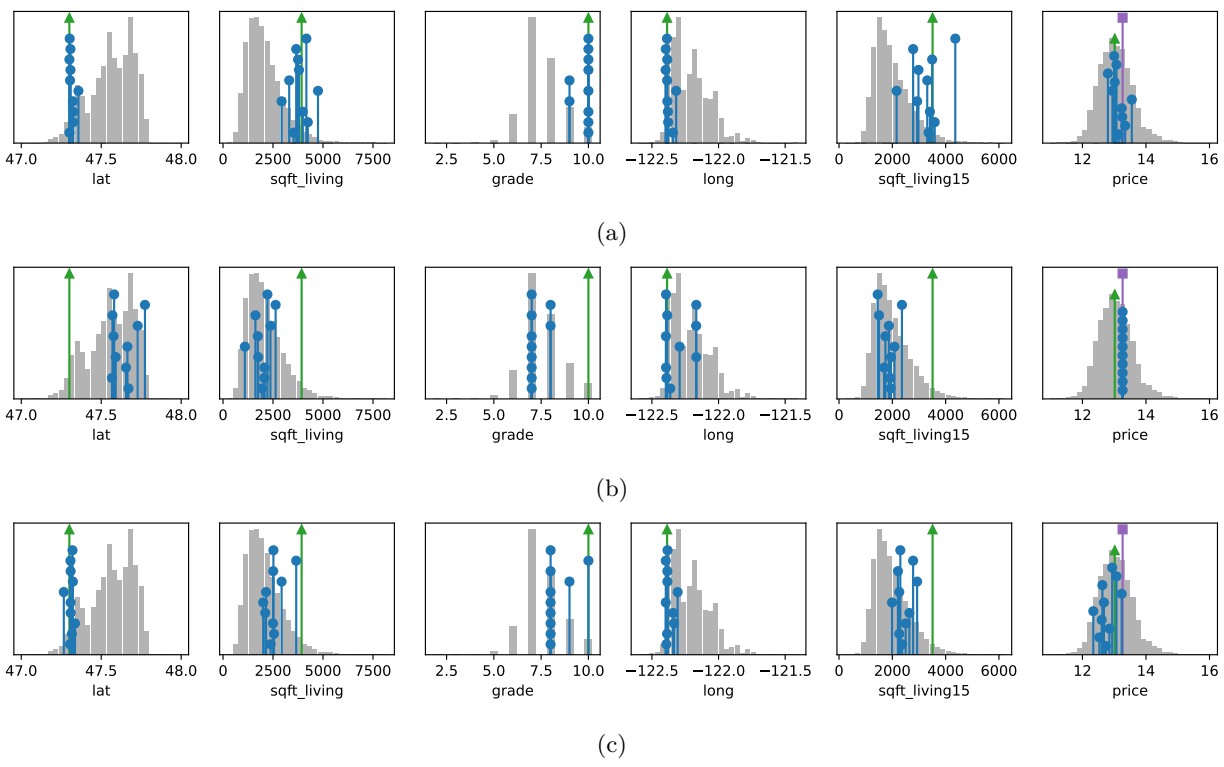

Figure 2: Members of the training dataset (blue disks) similar to a testing sample (green arrow). Most important features and responses (right, predicted as a purple square), histogram of the training features and responses in the dataset in grey. a) Proposed measure of similarity $d$, b) increasing $|\hat{y} - y_n|$, c) increasing $\|\mathbf{x} - \mathbf{x}_n\|_\Sigma$

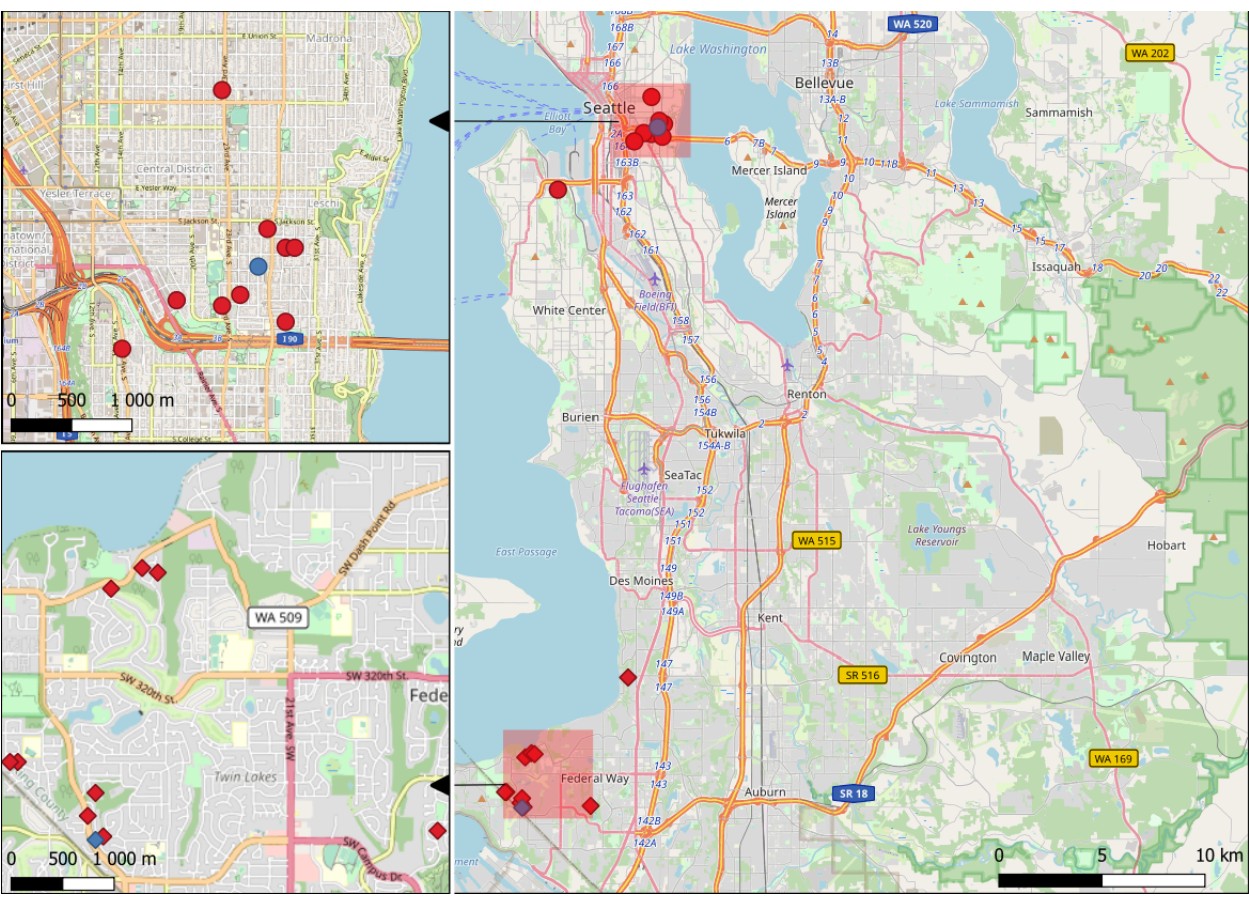

Figure 3: Map of King County. Two testing samples with the exact same selling price are represented in blue. Their respective similar samples are in red.

| Dataset | ExpGB | KNN ExpGB | KNN Euclidean | Variance |
|---------|-------|-----------|---------------|----------|
| *Abalone* | 4.33e0 | 4.81e0 | **4.63e0** | 9.74e0 |
| *Ailerons* | 2.26e-8 | **2.48e-8** | 4.10e-8 | 1.57e-7 |
| *Bike_sharing* | 1.40e3 | **1.69e3** | 1.26e4 | 3.33e4 |
| *Brazilian_houses* | 1.29e-1 | **1.40e-1** | 1.77e-1 | 6.18e-1 |
| *CPU_act* | 4.76e0 | **5.63e0** | 2.17e1 | 3.20e2 |
| *Diamonds* | 2.67e5 | **3.14e5** | 7.23e5 | 1.28e6 |
| *Elevators* | 4.01e-6 | **5.83e-6** | 1.29e-5 | 1.28e-5 |
| *Houses* | 4.54e-2 | **4.76e-2** | 7.90e-2 | 3.23e-1 |
| *House_16h* | 3.59e-1 | **4.08e-1** | 4.65e-1 | 8.34e-1 |
| *House_sales* | 2.94e-2 | **3.31e-2** | 5.26e-2 | 2.85e-1 |
| *Medical_charges* | 5.94e-3 | 6.51e-3 | **6.48e-3** | 3.20e-1 |
| *Miami_housing_2016* | 1.94e-2 | **2.22e-2** | 3.87e-2 | 3.17e-1 |
| *NYC_taxi_green_dec_2016* | 6.65e0 | 8.24e0 | **6.78e0** | 7.17e0 |
| *Pol* | 1.71e1 | **2.36e1** | 8.78e1 | 1.72e3 |
| *Sulfur* | 1.13e-3 | 1.35e-3 | **9.83e-4** | 2.70e-3 |
| *Superconductor* | 9.03e1 | **9.41e1** | 1.13e2 | 1.19e3 |
| *Wine* | 3.79e-1 | **4.20e-1** | 4.89e-1 | 7.58e-1 |
| *Yprop_4_1* | 7.27e-4 | 7.71e-4 | **7.46e-4** | 8.02e-4 |

Table 3: MSE of the KNN estimator compared to ExpGB and the variance of the response. For each dataset, the best performance between KNN EpxGB and KNN Euclidean is in bold.

this KNN Euclidean is obtained via cross-validation for all datasets. MSE are higher for KNN ExpGB than ExpGB, but not significantly so, showing that the $K$ prototypes have similar responses to the testing sample.

### 4.5.2 Features of prototypes

In this section, we assess the capability of our algorithm to extract meaningful prototypes in terms of features. To do so, we compute an estimation of each feature of a tested sample by averaging the features of its prototypes as selected in section 3.1. We then compare the MSE of this estimation with the variance of each feature, the variance corresponding to the MSE that is made by the naive estimator consisting of the mean of the full dataset. In table 4, the ratio between the MSE of the estimator defined above (KNN ExpGB) and the variance of the features are given. We can see that the features of the closest training samples are concentrated near the features of the testing sample.

### 4.5.3 Stability of prototype sets over iterations

At each iteration, the set of prototypes for a given example may change since all weights are updated. A desirable property for the proposed approach is a certain stability of the sets of prototypes. Figure 4 shows, at each iteration, the averaged cardinality of the symmetric difference of the sets of prototypes before and after an iteration. As the algorithm progresses, changes in the sets of prototypes become less probable, and this is true even for $T$ relatively small. We have only represented the case for one dataset here, but we observe this phenomenon no matter the dataset tested.

### 4.6 Overfitting and decomposition weights

Decomposing weights of a prediction of the response of a training sample are given in Figure 5 both classical gradient boosting (catboost) and ExpGB. As pointed out, decomposition weights can be negative in gradient boosting, and overfitting is visible here as the weight associated with the training sample tends to 1, while the other weights are decaying to 0. In contrast, decomposition weights obtained by ExpGB are spread out and in this particular case, do not involve the training sample considered here.

| Dataset | Feat 1 | Feat 2 | Feat 3 | Feat4 |
|---|---|---|---|---|
| *Abalone* | 1.48e-2 | 1.01e-2 | 1.61e-3 | 2.47e-1 |
| *Ailerons* | 1.65e-2 | 4.97e-2 | 5.94e-2 | 1.32e-1 |
| *Bike_sharing* | 6.16e-3 | 1.37e-1 | 7.39e-2 | 3.76e-2 |
| *Brazilian_houses* | 7.02e-1 | 7.30e-2 | 9.95e-2 | 3.02e-2 |
| *CPU_act* | 3.52e-1 | 8.20e-2 | 1.17e-2 | 4.84e-2 |
| *Diamonds* | 1.99e-3 | 2.24e-3 | 9.95e-4 | 1.5e-3 |
| *Elevators* | 1.42e-2 | 7.48e-3 | 2.66e-2 | 7.84e-3 |
| *Houses* | 4.57e-3 | 5.05e-3 | 3.41e-2 | 2.36e-1 |
| *House_16h* | 1.65e0 | 1.47e-1 | 1.14e-1 | 1.41e-1 |
| *House_sales* | 8.30e-3 | 7.31e-2 | 6.72e-2 | 1.18e-2 |
| *Medical_charges* | 3.89e-4 | 4.80e-2 | 1.96e-1 | X |
| *Miami_housing_2016* | 4.26e-2 | 4.09e-3 | 9.57e-3 | 1.15e-2 |
| *NYC_taxi_green_dec_2016* | 1.92e-3 | 0.00 | 1.27e-5 | 2.67e-2 |
| *Pol* | 4.41e-2 | 8.89e-2 | 11.3e-1 | 2.47e-1 |
| *Sulfur* | 4.22e-2 | 5.26e-2 | 2.36e-2 | 5.89e-2 |
| *Superconductor* | 1.78e-2 | 1.09e-2 | 1.07e-1 | 3.42e-1 |
| *Wine* | 7.97e-2 | 1.30e-1 | 1.34e-1 | 2.51e-1 |
| *Yprop_4_1* | 1.03e-1 | 3.09e-1 | 1.70e-1 | 1.13e-1 |

Table 4: Ratio between the MSE of the estimation of the features by KNN ExpGB and the variance of the features. Features are ordered by decreasing importance.

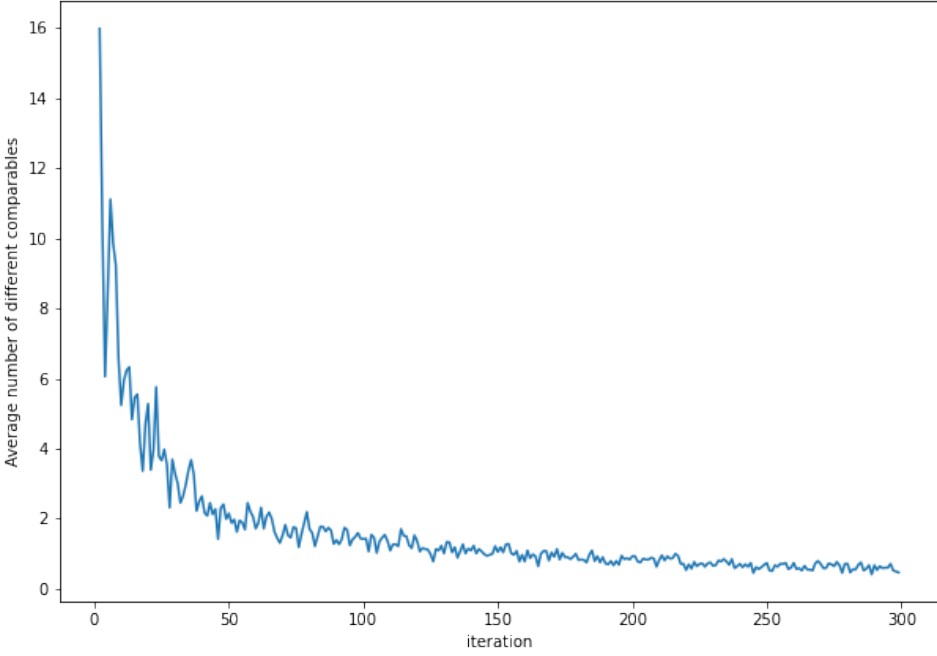

Figure 4: Average cardinality of the symmetric difference between prototype sets at successive iterations.

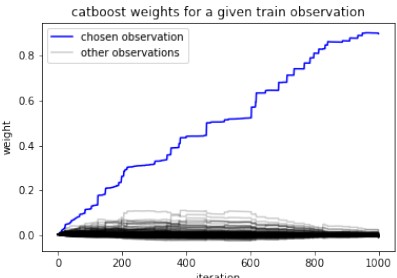 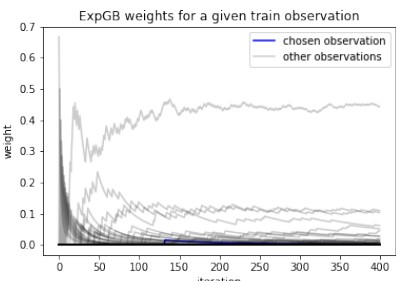

Figure 5: Bike sharing dataset. Decomposition weights $w_n^t(\mathbf{x})$ for a sample $\mathbf{x}_m$ in the training set, in the function of the number of iterations $t$. Top: Gradient boosting (catboost). Bottom: Proposed algorithm. The weights $w_m^t(\mathbf{x}_m)$ of the response of $\mathbf{x}_m$ in its prediction are highlighted.

| Dataset | Cat | XGB | Sklearn | ExpGB |
|---|---|---|---|---|
| *Abalone* | 1.07e+01 | 1.58e+01 | 3.16e+01 | **1.03e+01** |
| *Ailerons* | **4.10e-08** | 5.88e-08 | 1.25e-07 | 4.16e-08 |
| *Bike_sharing_demand* | 4.09e+03 | 5.96e+03 | **4.00e+03** | 4.38e+03 |
| *Brazilian_houses* | **1.93e+00** | 1.30e+01 | 1.66e+01 | 2.16e+00 |
| *CPU_act* | 1.34e+02 | 5.16e+02 | 4.43e+02 | **1.01e+02** |
| *Diamonds* | **7.82e+05** | 2.64e+06 | 2.23e+06 | 1.09e+06 |
| *Elevators* | **1.46e-05** | 3.24e-05 | 4.42e-05 | **1.46e-05** |
| *Houses* | 1.50e+00 | 5.89e+00 | 1.01e+01 | **9.51e-01** |
| *House_16h* | 1.53e+00 | 4.44e+00 | 6.72e+00 | **1.20e+00** |
| *House_sales* | **2.67e+00** | 1.50e+00 | 7.34e+00 | 1.50e+01 |
| *Medical_charges* | **1.90e-01** | 5.32e-01 | 1.44e+00 | 3.42e-01 |
| *Miami_housing_2016* | 1.39e+00 | 7.45e+00 | 9.08e+00 | **1.25e+00** |
| *NYC_taxi_green_dec_2016* | 9.22e+00 | 6.98e+00 | **6.55e+00** | 7.75e+00 |
| *Pol* | 1.63e+02 | 1.99e+02 | 3.59e+02 | **1.02e+02** |
| *Sulfur* | **1.51e-03** | 2.01e-03 | 1.83e-03 | 1.53e-03 |
| *Superconductor* | 1.60e+02 | 1.94e+02 | 2.37e+02 | **1.59e+02** |
| *Wine* | 3.06e+00 | 5.83e+00 | 6.16e+00 | **2.13e+00** |
| *Yprop_4_1* | 2.67e-02 | 6.73e-02 | 9.50e-02 | **8.59e-03** |

Table 5: Errors (MSE) on the test set for each of the 18 datasets. We modified the target values on approximately 1% of the data to introduce outliers. The hyperparameters are determined on this new dataset using cross-validation. Each model is trained using a $\ell_2$ loss.

### 4.7 Sensibility to outliers

From the definition of the proposed algorithm, one can see that a prediction is mainly based on the extreme points of the data. To ensure that ExpGB does not heavily depend on outliers in the dataset, we propose to look at the predictive performance of each gradient-boosting algorithm when the dataset contains a lot of outliers. For this purpose, we randomly modified the target features of 1% of the observations in each dataset. Half of them will have their target values divided by 10, and the other half will be multiplied by 10. We then computed the best hyper-parameters in this case for every model using the same method described in 4.2.1. Each model has been trained using a $\ell_2$ and a $\ell_1$ loss, results are presented respectively in table 5 and 6. As expected, we can see that all algorithms have their error improved, with those trained with a $\ell_1$ loss being more robust to outliers. Interestingly, whether it is for $\ell_1$ or $\ell_2$ loss function, ExpGB achieves similar performances as the other algorithms, proving that ExpGB is as robust to outliers as the other algorithms.

| Dataset | Cat | XGB | Sklearn | ExpGB |
|---|---|---|---|---|
| *Abalone* | 5.02e+00 | 4.73e+00 | **4.38e+00** | 4.92e+00 |
| *Ailerons* | 2.44e-08 | **2.43e-08** | 3.88e-08 | 2.67e-08 |
| *Bike_sharing_demand* | 1.75e+03 | 2.08e+03 | 1.89e+03 | **1.69e+03** |
| *Brazilian_houses* | **1.32e-01** | 1.34e-01 | 1.36e+01 | 1.78e-01 |
| *CPU_act* | **9.59e+00** | 4.54e+01 | 6.96e+01 | 1.69e+01 |
| *Diamonds* | **3.28e+05** | 3.35e+05 | 3.59e+05 | 3.67e+05 |
| *Elevators* | 5.17e-06 | 5.96e-06 | 5.61e-06 | **4.98e-06** |
| *Houses* | 5.83e-02 | **5.51e-02** | 7.55e-02 | 2.06e-01 |
| *House_16h* | 4.31e-01 | 4.13e-01 | **3.63e-01** | 7.85e-01 |
| *House_sales* | **1.07e-01** | 1.75e-01 | 3.12e-01 | 1.20e-01 |
| *Medical_charges* | **7.14e-03** | 9.00e-03 | 1.99e-02 | 2.59e-02 |
| *Miami_housing_2016* | **2.61e-02** | 1.98e-01 | 2.66e-02 | 1.58e-01 |
| *NYC_taxi_green_dec_2016* | 6.80e+00 | 6.73e+00 | 6.73e+00 | **6.71e+00** |
| *Pol* | 3.22e+01 | 2.75e+01 | 2.43e+01 | **2.11e+01** |
| *Sulfur* | **1.07e-03** | 1.38e-03 | 1.64e-03 | 1.21e-03 |
| *Superconductor* | 9.33e+01 | 9.56e+01 | 9.95e+01 | **9.10e+01** |
| *Wine* | 5.63e-01 | 5.42e-01 | 4.87e-01 | **4.81e-01** |
| *Yprop_4_1* | **1.04e-03** | 1.68e-03 | 1.09e-03 | 1.67e-03 |

Table 6: Errors (MSE) on the test set for each of the 18 datasets. We modified the target values on approximately 1% of the data to introduce outliers. The hyperparameters are determined on this new dataset using cross-validation. Each model is trained using a $\ell_1$ loss.

## 5 Conclusion and future works

In this article, we showed that for regression tasks with an $\ell_2$ loss, the predictions of GBDT can be written as linear combinations of the training values, with weights summing to one, depending on the sample of interest. A new tree-based gradient boosting algorithm based on the Frank-Wolfe algorithm was proposed, constraining the linear combinations to be convex and applicable to a wider range of losses. Results on several datasets showed that performances are comparable to state-of-the-art gradient-boosting implementations.

We have shown that similar (in terms of features and response) data can be identified by selecting training samples with low $l_1$-norm between their weights and the weights of the tested samples.

Such results open the way for many developments. Concerning the ExpGB algorithm, some original elements for proving at a theoretical level the algorithm convergence will be developed in future works. Indeed, mimicking the Franck-Wolfe proof does not allow direct proof of the convergence. Another important perspective would be to exploit the outputs of the proposed algorithm in a clustering perspective. Indeed, each weight vector of a predicted observation corresponds to the observation "signature" and can be used in any similarity measures. Finally, implementations of our algorithm for classification tasks are straightforward by simply changing the loss function.

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
