# OpenReview forum: "Enhancing the Explainability of Gradient Boosting for Regression Problems through Comparable Samples Selection"
_TMLR — Rejected by TMLR_

### Review · Reviewer_Poza · 2024-07-03

**Summary Of Contributions:**

The paper discusses Gradient-boosted Decision Trees (GBDT), a powerful method for classification and regression, but highlights its lack of explainability. To address this, the authors propose identifying "comparable samples" from the training data that heavily influence specific predictions. They demonstrate that GBDT predictions can be expressed as weighted sums of training data when using certain loss functions, with the caveat that weights can be negative. This weight behavior suggests overfitting issues. To counter this, they introduce Explainable Gradient Boosting (ExpGB), which imposes nonnegativity constraints on the weights and uses a Frank-Wolfe algorithm-inspired method instead of gradient descent. ExpGB's predictions are interpretable as convex combinations of training targets, facilitating the identification of similar training samples. Comparative analyses on various datasets show that ExpGB enhances explainability without sacrificing prediction quality.

**Audience:**

Yes

**Claims And Evidence:**

Yes

**Requested Changes:**

Please address the points raised in the "Weaknesses"

**Strengths And Weaknesses:**

Strengths

- The topic discussed in the paper is interesting and important
- The proposed method appears to be novel

Weaknesses

- The authors focus on a variant of GBDT for regression problems. Why not develop an explainable approach for the original GBDT? This gives reviewers the impression that the authors aim to alter the original method to make it explainable rather than providing explainability for the original problem. As a result, the proposed method loses its appeal because the authors seem to change the problem.

- Additionally, demonstrating the proposed method on a classification problem is important

- Related works should be updated and compared, e.g.,
Delgado-Panadero, Ángel, et al. "Implementing local-explainability in gradient boosting trees: feature contribution." Information Sciences 589 (2022): 199-212.

- The author should also compare their method with SHAP for GBDT because SHAP is a model-agnostic method. While SHAP has its own computational issues, several approaches have been proposed in the literature of explainable AI to approximate SHAP values.

- There are several explainable deep learning approaches. Could the authors conduct comparisons between those models and the proposed model on the same classification or regression task?

- Code should be provided for reproducibility.

---

> ### Author Response · Authors · 2024-08-05
> **Response to Reviewer Poza**
>
> We wish to thank the reviewer for their time and comment on our work. We would like to make the following comments in relation to their requested changes:
> > The authors focus on a variant of GBDT for regression problems. Why not develop an explainable approach for the original GBDT? This gives reviewers the impression that the authors aim to alter the original method to make it explainable rather than providing explainability for the original problem. As a result, the proposed method loses its appeal because the authors seem to change the problem.
>
> We have indeed chosen to follow an approach where we modified the algorithm to makes its interpretability in easier (in the sense of selection training samples comparable to a given test sample). However, the regression problem remains the same, and the proposed algorithm shares important characteristics with the original gradient boosting algorithm: its prediction are obtained by combining weak learners in the form of decision trees, and its performances are similar to reference implementations of GBDT. This, in our opinion, makes its a valid replacement of GBDT when comparable samples are useful.
>
> > Additionally, demonstrating the proposed method on a classification problem is important
>
> The present submission is focused on regression problems. We agree that an extension to classification problems is valuable. This extension has been investigated and submitted to another venue. The problems and methods being different between the two submissions, we note that there is no overlap in figures, text, or results between them.
>
> > Related works should be updated and compared, e.g., Delgado-Panadero, Ángel, et al. "Implementing local-explainability in gradient boosting trees: feature contribution." Information Sciences 589 (2022): 199-212.
> > The author should also compare their method with SHAP for GBDT because SHAP is a model-agnostic method. While SHAP has its own computational issues, several approaches have been proposed in the literature of explainable AI to approximate SHAP values.
>
> Both the cited article and Shap are feature-based algorithms that improve the explainability of a model, while our method is an example-based method. This means that the former methods aim to quantify the impact of the test’s features on the prediction, whereas our aim is to quantify which samples are similar to the tested samples. The two approaches are fundamentally different, and we are not aware of any ways to compare these two kinds of explainability methods.
>
> > There are several explainable deep learning approaches. Could the authors conduct comparisons between those models and the proposed model on the same classification or regression task?
>
> As stated in Grinsztajn et al., Why do tree-based models still outperform deep learning on typical tabular data?. Advances in Neural Information Processing Systems, 2002, , deep learning approaches does not compare well with gradient boosting method regarding tabular data. Moreover, the same issue applies with Weaknesses 3 and 4, that is the explainable method are feature based and not example based. This is why we did not include a comparison with deep learning methods.
>
> > Code should be provided for reproducibility.
>
> We submitted the source code with an example on the bike-sharing dataset, including the code to generate all graphics and metrics used with this dataset.

---

### Review · Reviewer_dgD7 · 2024-07-18

**Summary Of Contributions:**

This paper propose a variant of gradient boosting algorithm, called Explainable Gradient Boosting (ExpGB), which adds non-negativity constraints on the weights and substitute gradient descent with a methodology inspired by the Frank-Wolfe algorithm. The resulting algorithm has not only better explainability because of the non-negativity of the weights, but also smaller testing error because of the regularization effect of the constraints.

**Audience:**

Yes

**Broader Impact Concerns:**

I didn't find any concerns on the ethical implications of this work.

**Claims And Evidence:**

Yes

**Requested Changes:**

No significant changes are requested.
But I do wonder whether it is possible to extend this work to classification problems and cross-entropy loss.

**Strengths And Weaknesses:**

Strengths:
1. This paper propose a novel variant of gradient boosting algorithm, called Explainable Gradient Boosting (ExpGB), which adds non-negativity constraints on the weights and substitute gradient descent with a methodology inspired by the Frank-Wolfe algorithm.
2. The proposed algorithm has better explainability because of the non-negativity of the weights.
3. The proposed algorithm has smaller testing error because of the regularization effect of the constraints.

Weaknesses:
1. The proposed algorithm is limited to regression problems with L2 loss, while GBDT is typically used more frequently in classification problems.

---

> ### Author Response · Authors · 2024-08-05
> **Response to Reviewer dgD7**
>
> Thank you for your comment and your positive feedback on our work. Here are some additional remarks:
>
> > The proposed algorithm is limited to regression problems with L2 loss, while GBDT is typically used more frequently in classification problems.
>
> The submission is indeed limited to regression problems. However, while the discussion of gradient boosting in section 2 is limited to the L2 loss, the proposed algorithm can be applied to more general losses, such as the L1 loss used in section 4.7. The text has been revised to clarify this point.
>
> > No significant changes are requested. But I do wonder whether it is possible to extend this work to classification problems and cross-entropy loss.
>
> As the proposed method is not limited to the $\ell_2$ norm, a variant adapted to classification problems has indeed been investigated, and submitted to another venue. The problems and methods being different between the two submissions, we note that there is no overlap in figures, text, results  between them.

---

### Review · Reviewer_Cmz4 · 2024-07-24

**Summary Of Contributions:**

This paper aims to improve the explainability of Gradient-boosted decision Trees
(GBDT). It introduces explainable gradient boosting (ExpGB) by imposing
nonnegativity constraints on the weights and replacing gradient descent with a
Frank-Wolfe type algorithm. The claimed explainability is based on the fact that the resultant predictions from the proposed algorithm are linear functions of the training targets. Multiple data sets are used to illustrate the proposed algorithm. There are no theoretical investigations.

**Audience:**

Yes

**Claims And Evidence:**

No

**Requested Changes:**

1. Improve the quality of presentation and define all notations clearly before
   or right after their usage, e.g., the definition in (3) is unclear; the
   scalar product in (12) should be defined before or right after its
   statement.
2. Shorten the paper by removing unnecessary repetitions, e.g., section 1.1 is
   mainly repeating previous texts; the beginning of section 3 is repeating the
   end of section 2.
3. Consider general loss functions and predictors, not only squared loss. I am
   also confused by the statement that the proposed algorithm is a linear
   estimator. According to the literature, the best linear unbiased estimator is
   the least squares estimator. Why people would want to use the GBDT? Does this
   mean the model in consideration is just a linear model and the proposed
   algorithm just gives the least squares estimator under the squared loss?
4. It seems the γ in Algorithm 1 depends on the response y. If this is the case,
   the algorithm does not produce a linear estimator.
5. According to some statements, e.g., "the weights $w_n^t(x)$ are increased
   only for training samples $x_n$ falling in the same leaf as the tested
   sample" seems to indicate that the training algorithm depends on the test
   sample. This needs to be clarified.

**Strengths And Weaknesses:**

Strength:
1. The topic on explainability of GBDT is interesting and important.
2. Multiple data sets are used to illustrate the proposed algorithm.

Weakness:
1. The paper lacks theoretical investigations.
2. The overall quality of presentation is low and needs significant
   improvement. The paper is hard to follow and the contribution is not well
   explained. There are multiple repetitions making the paper unnecessarily
   long. Some key notations are not discussed clearly enough; for example,
   scalar product makes the computation much harder (if not impossible) and the
   density of the feature is never know and difficult to obtain in
   practice. There is not discussion on these important points, and it is
   difficult to connect these definitions with the corresponding algorithms
   (Algorithm 3, if I understand correctly).
3. The paper is titled with "Enhancing the Explainability", but the paper does
   not provide any explanation on how the proposed algorithm can enhance the
   explainability of GBDT. The restricted focus on squared loss and linear
   predictor makes focus of the investigation rather narrow.

---

> ### Author Response · Authors · 2024-08-05
> **Response to Reviewer Cmz4**
>
> We thank the reviewer for the evaluation of our work. We address their concerns below:
>
> > Improve the quality of presentation and define all notations clearly before or right after their usage, e.g., the definition in (3) is unclear; the scalar product in (12) should be defined before or right after its statement.
>
> Revisions have been made to clarify notations and definitions, particularly for (3) and (12). The issue with the unknown density $p$ has been clarified, as it is not needed to solve the optimization problem (13).
>
> > Shorten the paper by removing unnecessary repetitions, e.g., section 1.1 is mainly repeating previous texts; the beginning of section 3 is repeating the end of section 2.
>
> The beginning of section 3 has been revised to minimize overlap with the end of section 2.
>
> > Consider general loss functions and predictors, not only squared loss. I am also confused by the statement that the proposed algorithm is a linear estimator. According to the literature, the best linear unbiased estimator is the least squares estimator. Why people would want to use the GBDT? Does this mean the model in consideration is just a linear model and the proposed algorithm just gives the least squares estimator under the squared loss?
>
> While the discussion of gradient boosting in section 2 is limited to the L2 loss, the proposed algorithm can be applied to more general losses, such as the L1 loss used in section 4.7. The text has been revised to clarify this point.
>
> We remark that we do not claim that the proposed algorithm is a linear estimator, but that for a given sample, its prediction can be written as a linear combination of the training data, with weights function of the tested sample.
>
>
> > It seems the γ in Algorithm 1 depends on the response y. If this is the case, the algorithm does not produce a linear estimator.
>
> As pointed out above, algorithm 1 is not expected to be linear.
>
>
> > According to some statements, e.g., "the weights are increased only for training samples falling in the same leaf as the tested sample" seems to indicate that the training algorithm depends on the test sample. This needs to be clarified.
>
> The training step of the algorithm does not depend on the tested dataset. However, the computation of the weight $w$ associated with a prediction for a given tested sample is made in the predicting step, which will depend on the considered tested sample. The estimator being non-linear, the weights are allowed and are supposed to depend on the tested sample.

---

### Decision · Action_Editor_HeTp · 2024-08-27

**Recommendation:** Reject

**Comment:**

The reviewers were mostly negative towards the quality of the paper.
While the authors addressed some concerns raised by the reviewers, such as the applicability to classification problems and questions about linearity, one critical issue remains unresolved.
This issue concerns whether the proposed method is truly superior to other example-based explanation methods.
This point is closely related to Reviewer Cmz4's question about "Enhancing the Explainability" and Reviewer Poza's remark that "There are several explainable deep learning approaches."

Although the authors claimed that their method falls under example-based explanations, there is no discussion nor comparison with other example-based methods in the paper.
This lack is crucial and significantly impacts the evaluation of the paper.

One representative example of an example-based explanation method is Influence Function (IF) [Ref1].
There are some studies on IF applied to GBDT [Ref2,3].
Additionally, in the current paper, the authors focused mostly on using example weights for similarity search.
A natural question then arises: is the proposed similarity better than the existing ones?
For tree ensembles, similarity metrics such as RandomForest Kernel [Ref4] would be relevant.
An important implication here is that there are numerous example-based explanation methods and similarity metrics.
A proper evaluation of the novelty and effectiveness of the proposed method over these existing methods is essential.

* [Ref1] Understanding black-box predictions via influence functions, ICML17.
* [Ref2] Finding influential training samples for gradient boosted decision trees, ICML18.
* [Ref3] Adapting and Evaluating Influence-Estimation Methods for Gradient-Boosted Decision Trees, JMLR23.
* [Ref4] The Random Forest Kernel and creating other kernels for big data from random partitions, arXiv14.

**Audience:**

This paper proposes a new example-based explanation method for Gradient-Boosted Decision Trees (GBDT).
Model explainability is an important research topic in machine learning and falls within the scope of TMLR.

**Claims And Evidence:**

This paper presents the detailed methodology and experimental results.
However, despite the method discussed belonging to the category of example-based explanations, it lacks references to and comparisons with related research in this area.
This lack is crucial and significantly impacts the evaluation of the paper.

**Resubmission Of Major Revision:**

The authors may consider submitting a major revision at a later time.